# Consumers’ Preferences for the Traceability Information of Seafood Safety

**DOI:** 10.3390/foods11121675

**Published:** 2022-06-07

**Authors:** Mohammed Ziaul Hoque, Nazmoon Akhter, Mohammad Shafiur Rahman Chowdhury

**Affiliations:** 1School of Business and Economics in Tromsø, UiT The Arctic University of Norway, 9010 Tromso, Norway; 2Faculty of Business Administration, University of Chittagong, Chittagong 4331, Bangladesh; shafiur@cu.ac.bd; 3Department of Business Administration, BGC Trust University Bangladesh, Chattogram 4381, Bangladesh; akhternazmoon@gmail.com

**Keywords:** seafood traceability, food safety, chilled fish, emerging market, consumer preference, willingness to pay, Bangladesh

## Abstract

Due to importing food and the perpetual changes from conventional wet markets to supermarkets in emerging markets, consumers have the opportunity to base their buying decisions on traceability systems. Seafood traceability systems involve information on production mode, inspection notes, sustainable sources, and sources of origin to provide consumer protection and help ensure that all seafood is safe to consume. This study aims to explore seafood markets by assessing the demand for traceability information attributes by utilising data from an experimental survey in an emerging market such as Bangladesh. The data were analysed using descriptive statistics, exploratory factor analysis, and a conditional logit model. The results demonstrate that consumers are concerned regarding vitamins, cholesterol, and preservatives, while they are little concerned about microbiological contamination, pesticide residues, genetic modification, and additives or artificial colours. The difference between the mean willingness to pay (WTP) for traditional and sustainable farmed fish is higher than that between the mean WTP for conventional and sustainable wild fish. In a ranked-choice voting system, the ‘production mode’ and ‘claim of safety control (e.g., being formalin-free)’ were the first and second most influential attributes in fish choices. The outcomes of the econometric model revealed that consumers are more likely to prefer traceability information about fish control (e.g., formalin-free), and they want to pay a price premium for this information. Alternatively, consumers are less likely to prefer farmed and imported fish, and their WTP for these fishes are highly inflated. This finding may be because consumers use wild and local origin as a cue for food safety or quality. This study hopes that the effects of such traceability information will optimise the production process and supply chain and help make seafood recall management more effective.

## 1. Introduction

As demanded, the consumption of safe and quality food is essential in people’s everyday lives to provide the energy and nutrients required to sustain life. The practice of consuming safe food not only saves lives and improves peoples’ health, but also ensures a country’s economic growth [1]. However, in many economies, the ‘food safety’ concept is hidden and problems are frequently overlooked [2], which can lead to alarming events in the food industry worldwide. Examples of these are mad cow disease in the United Kingdom in 1996; dioxin contamination in Belgium in 1996; the Escherichia coli outbreak in the United States in 2006; melamine-tainted milk powder in China in 2008 [3]; and the outbreak of Enteropathogenic Escherichia coli associated with contaminated fenugreek sprouts in Germany in 2011 [4]. Unsafe food practices have also increased due to the increased demand for food in the increased population worldwide. As food traceability relates to where and how foods are produced, it has the potential to be developed as a tool for providing information to consumers [5], which may reduce unhealthy practices and the risk of food fraud [6] and enhance the level of food safety. Traceability can also provide a foundation for addressing many of today’s food system issues, both in developed and developing economies.

To meet the demands of approximately 10 billion people by 2050, it is planned to increase global food supply by 70% [7]. Such an increased demand creates stress in seafood produced by fish (wild vs. farmed), as consumers’ health awareness increases their fish consumption [8]. In recent years, fish eating has been appreciated because fish protein includes a high amount of fatty acids and less saturated fat than red meats [9]. Consuming fish protein reduces the risk of cardiovascular disease, and the omega-3 fatty acids involved in fish consumption are vital for neurological improvement and general health [10]. In addition, fish consumption meets the heath demand for calcium, selenium, and zinc [11]. These increased demands have led to overfishing, becoming a matter of alarm because of the depletion of natural fish stocks [12]. Consequently, the efficient role of fishery management is required to protect marine biodiversity and ensure their contribution to food security [13].

Advanced countries place a significant focus on food safety and security, as well as on the sustainable fish production process. They try to contribute to fish welfare and safe fish consumption because environmental concerns and ecological sustainability influence consumers’ perceptions [14]. Again, consumers’ purchase decisions on fish are mostly affected by taste, health, and nutritional factors [15]. In the USA’s Pacific Northwest, peoples’ preferences for seafood have been shown to be influenced by health, environment, familiarity, and price [16]. Generally, most consumers perceive wild fish to be better in quality compared to farmed fish. Belgian consumers ranked wild fish as healthier, better tasting, and more nutritious than farmed fish [17]. Although wild fish are higher in quality and fresh, with fewer antibiotics used in their production, farmed fish are widely available at a lower price [18]. Moreover, French consumers found that farmed fish played a positive role in fish welfare and environmental sustainability [19]. However, such safety and sustainability issues have not been paid full attention in emerging markets and have been mostly unexplored with regard to the consumers of farmed fish [20]. Furthermore, due to limited knowledge about the food production process, consumers cannot make their purchasing decisions effectively [21].

In buying decisions, traceability systems are gaining importance as assurance mechanisms for food safety and quality to regain consumer confidence [22]. A traceability approach can transmit information throughout the supply chain more efficiently, ensuring sanitary security and the information required by consumers [23]. Food traceability systems increase food safety, health, naturalness, quality, trust, control guarantees, and environmental protection [24]. The impact of traceability is immediate for goods entering developed markets. In many low- and middle-income countries, traceability techniques offer a realistic solution to overcoming poorly functioning supply chains [25]. Therefore, food traceability is necessary to ensure food safety and security, and it has been accepted in developing and emerging markets because of the high levels of fraudulent products entering the markets and the rise of a growing consumer middle class [26]. Such fraudulent products and the series of food scandals have eroded consumer confidence and led to changes in food safety systems [27]. In these dynamic systems, the baseline for food safety has also improved in emerging markets. In an emerging South Asia market, Bangladesh, government authorities have adopted integrated food safety control [28]. However, in local markets, food is contaminated by the unsafe chemicals used in diverse food supply chain stages from farms to consumers [29]. As unsafe food practices threaten consumers’ health and increase healthcare expenditures [1], two incidences of food scandals in Bangladesh in 2008 and 2016 have made consumers become more conscious about food safety and quality [30].

The high financial level of fish and fisheries products, estimated by the FAO at around USD 60 billion per year, could attract unscrupulous producers and traders to practise the unethical selling of fish products with false authenticity values [31]. In this context, research on general and specific fish product traceability systems which verify existing papers on traceability systems, production methods, and the geographical origin of fish would be beneficial [31]. Although consumers’ perceptions of food traceability have been previously examined, little is known about their food-safety concerns and their preference for traceability information of seafood safety, more specifically, in emerging markets. Therefore, this study aims to fill this knowledge gap. To achieve its objective, the study examines consumers’ food safety concerns and preferences towards traceable fish attributes (production method, geographical origin, safety claim, and price) in an emerging market, such as Bangladesh. The willingness of consumers to pay for these attributes, and for overall safe fish, are also explored.

In Bangladesh, a common practice among food vendors is to spray fish with chemical preservatives, including formalin, while they are transported through the domestic market chain [29] to boost their lifespan and appearance [32]. In addition, formalin has been frequently reported at levels in excess of those recommended for addition as a preservative to fresh fish [29]. Therefore, this paper introduces a new food traceability attribute, the safety claim of being ‘formalin-free’, and the status quo ‘no safety claim’ to obtain consumers’ real insights into the traceability information of food safety. A sample of 404 consumers from the two major cities of Chittagong and Dhaka in Bangladesh were interviewed directly (face to face) in the experimental design. The collected data were analysed with a conditional logit (CNDL) model. We expect that the outcomes of the study such as consumers’ awareness of seafood safety and their willingness to pay for the traceability information of seafood safety could help in formulating government regulations for seafood traceability.

The composition of the study is as follows: The literature review and theoretical framework are first presented, followed by a discussion of the data and the empirical model. The research results are then discussed, and the paper ends with the concluding remarks and suggestions for directions for further research.

## 2. Literature Review

Presently, fish consumption has increased along with the increased population around the world [12]. In the case of fish consumption, consumers prefer domestic to foreign products [33] due to the short transportation distance from producer to consumer and the possibility to check on the production process [34]. For example, consumers in Germany, Denmark, and Poland have been shown to prefer smoked trout produced in their home country [34]. Moreover, consumer preference and the consumption of fish depend on several factors, such as colour, smell, taste, texture, convenience, health, availability, safety, price, ethical concerns, natural content, their socioeconomic background, food consumption patterns, and the media [35,36]. Consumers’ preferences and buying behaviours also vary based on the fish cultivation process (wild or farmed). They generally prefer wild fish due to its high quality compared to farmed fish [37]. For example, European consumers have a positive perception of wild fish [37]. However, wild fish are more expensive and luxurious than farmed fish [38].

Farmed fish are readily available at a lower price than wild fish [37]. Besides, the most crucial contribution of farmed fish in the food market is to fish procurement, being a valuable alternative to caught fish and helping to save wild fish from extinction [34,39]. It is also viewed that farmed fish provide safety, health, and sustainability. For instance, consumers in the northeast and mid-Atlantic regions of the USA have been shown to prefer farmed salmon where such fish are cultivated under strict control procedures to maintain higher quality and safety than wild salmon [40]. Additionally, European consumers who possess sufficient information about aquaculture related to food safety and sustainability perceive farmed fish as playing a positive role in their lives [41]. However, a question has arisen over its quality, as detrimental chemicals such as formalin are used to maintain long-lasting freshness [29]. Different artificial chemicals such as carbide, formalin, heavy metals, textile colours, artificial sweeteners, DDT, and urea are used in the food production process, harming human health [42]. Therefore, human health is threatened. For this reason, fish products need to be guaranteed to be safe for health [43]. In this case, food traceability will help trace the food products’ movement from their different production stages to their distribution to the end consumers [44] and help remove unsafe goods from food markets [3].

Presently, consumers’ concerns about personal health and food quality are closely related to the continuous advances in government regulations and traceability systems [45]. Traceability systems are relevant and help achieve the safety of supply chains as they provide health alerts by defining foodstuffs [46]. Seafood safety and traceability systems based on radio frequency identification, blockchains, wireless sensor networks, and the Internet of Things (IoT) give reliability from farm to fork [47]. However, consumers have limited knowledge about traceability information [48], usually perceiving it to concern food safety and ensure food quality [49]. Regarding food quality, information about a product’s origin can act as a signal to build consumers’ confidence [50]. Such information provides details of the product’s intrinsic features and attempts to ensure sustainability by monitoring and controlling the production process [46]. Therefore, food quality signals and food traceability are not the same; instead, they are interrelated, as food traceability intends to provide information about the origin of products, which is considered a food quality signal [46]. Therefore, food quality signals lead to food traceability, which is regarded as a relevant tool because it identifies and recognises various aspects of the production process [49]. To connect food traceability directly to end consumers, it is now imperative to provide food quality information [46]. In this case, labelling, product brand names, and shop assistants can help consumers to understand fish production-related information [46]. Labelling food products is a crucial tool to allow consumers to obtain food quality information, mainly when making a purchase decision [45].

Labelling is an effective method of providing messages in the form of claim and sources [51], affecting the persuasion process and building trust among consumers [52]. Using a label provides relevant information about nutrition, quality, and food safety, influencing consumers’ purchasing decisions [53]. Moreover, labelling also enhances products’ valuation, meaning consumers are more likely to pay extra for them [51]. For example, consumers in Germany prefer organic food and are prepared to pay a price premium for organic fish traceability [14]. According to Verbeke and Ward [54], older and female consumers are more influenced by guaranteed food quality schemes related to traceability. In contrast, young consumers have the least interest in the country of origin of the product [55]. Besides, highly educated females wish to obtain more information about fresh produce, whereas males with less education prefer to trace food-related information through the traceability systems of fresh produce [56]. Therefore, food traceability increases product prices [24], primarily through labelling [51]. In finding the optimum price at which to sell a food product, consumers’ willingness to pay (WTP) is a crucial factor.

Generally, the WTP for the availability of a traceability program is relatively high, and several studies report that consumers’ WTP mainly relies on their level of income, education level, and sensitiveness to food safety [57]. Consumers from different countries have different kinds of payment behaviours related to food traceability information [46]. For instance, those in France are prepared to pay a price premium for the implementation of traceability programs, whereas those in Spain and Portugal are not [46]. On the other hand, Spanish consumers are willing to pay a high price for food quality, but not for food traceability programs. Moreover, consumers in China are willing to pay 6% extra for fish with traceability than those without it [58]. In Taiwan, strong and effective possibilities for certified safe food exist as consumers are willing to pay high prices for milkfish and oysters produced under hazard analysis critical control point (HACCP) regulations [59].

Although advanced economies’ initiatives are appreciated in maintaining food safety, developing countries are still struggling against customer demand and, consequently, ignoring food safety issues [60]. In Bangladesh, an emerging Asian economy, foods are adulterated with various toxic artificial colours and harmful chemicals [61]; unauthorised food colours, formalin, and textile dyes are used in food, manufacturing, and processing in Bangladesh [62]. The formalin used in foods is currently a serious problem in Bangladesh, as supermarkets openly sell vegetables, fish, and fruit that have been treated with formalin to keep them fresh [63]. Scientists warn that formalin consumption directly through food can cause different types of cancer, especially lung cancer [64]. Though the use of formalin in food is prohibited according to the Safe Food Act–2013 (Section 23), unfortunately, formalin has been frequently reported in excess of the recommended levels, as it is added as a preservative to fresh fish by traders in Bangladeshi domestic markets [29].

In the Bangladesh domestic market, the mean levels of formalin of 402.35 mg/kg in imported fish and 118.60 mg/kg in local fish were much higher than the WHO recommended levels [29,32,65]. Therefore, the country’s fisheries sector has experienced limited expansion [66], and the negative effect of food adulteration is posing a significant threat to the population’s income and food security, with urges for immediate action by the government and policymakers [67]. In addition to domestic health issues, international demand, especially for seafood, e.g., Bangladeshi prawns, is also threatened due to the high rate of food contamination. EU buyers are rejecting many prawn consignments from Bangladesh because of the presence of banned nitrofuran and other hazardous chemicals [68], while asking for improved seafood safety procedures and trade traceability systems to ensure food safety [68].

The food traceability in Bangladesh is not up to the mark, even though food safety risks are high, given the inefficiencies in food transportation, handling, and storage. However, as the Bangladeshi economy continues to grow and consumers become more selective, there is likely to be increased focus on food safety and food origin [69]. Out of the international market’s food safety requirements, seafood traceability is a vital issue for Bangladeshi producers. Therefore, an adverse impact will occur if preventive action is not taken with regard to seafood safety by ensuring traceability in food production, processing, and marketing [70]. As part of seafood safety measures, with the government’s help, the Bangladesh Frozen Food Exporters Association introduced paper-based traceability for prawns in 2009. However, in terms of documentation and data analysis, the initiative was unsuccessful [68]. In January 2016, they introduced an e-traceability system. However, before launching such traceability projects, consumers’ perceived value of food traceability was not examined thoroughly. Therefore, this study aims to fill this knowledge gap, with consumers interviewed in a stated preference experimental design.

## 3. Data and Methods

### 3.1. Product and Participants

The Rui (*Labeo rohita*) is one of the most widely produced and consumed fish in Bangladesh [66]. Both wild-caught and farm-raised whole chilled Rui are available in Bangladeshi local markets. Additionally, whole chilled Rui from Myanmar and India are also found as imported fish in local markets. Therefore, this study only considers this whole Rui in its chilled form to show the impact of traceability information on consumers’ choice of fish. A structured questionnaire in the first language (Bengali) was presented to randomly selected households in a direct interview method to collect data. The two most economically and politically significant cities of Bangladesh, Dhaka and Chittagong, comprised the sample area (see Figure 1) [69]. Dhaka is the country capital, whereas Chittagong is a port (e.g., commercial) city.

People living in these two cities are relatively wealthy and maintain a high living standard. Additionally, these two cities’ fish consumption rates are higher than in other cities in Bangladesh [71]. That is why those living there are more conscious regarding food safety, and so they are most suitable for this investigation into consumers’ food safety concerns and their willingness to pay (WTP) for food traceability information (e.g., ‘formalin-free’ and ‘no safety claim’). We employed a stratified cluster sampling process in an attempt to gather the representative sample and consumers’ real insights. In Dhaka, respondents from both the north and south municipalities were recruited in equal proportions. Although Chittagong is not divided officially, it is also commonly separated into south and north from where respondents were recruited equally to ensure a representative sample. From the total of 404 samples, 203 respondents were recruited from Chittagong. Finally, the sampling distributions are as follows: Dhaka North (N = 101), Dhaka South (N = 100), Chittagong North (N = 102), and Chittagong South (N = 101).

In Bangladesh, in general, those in households those who are older than 20 are responsible for taking care of family food, and so people were recruited from this age group with a screening question asking if the participants were responsible for food purchase. The data collection was conducted from 12 January to 27 March 2019. A pre-test survey of 32 people from Dhaka and 30 from Chittagong was conducted to confirm that the respondents understood all the questions’ contents and that there were no semantic problems or linguistic complexities. As no significant restrictions were found, the survey was finalised, and was designed to take, on average, 15 min for each respondent.

### 3.2. Questionnaire and Measures

Three alternative fish options with four characteristics were considered to evaluate consumers’ heterogeneity in each choice. In the first section of the questionnaire, three choices were presented. Alternative ‘C’ was included to provide the possibility to choose neither alternative ‘A’ or ‘B’. The choices were organised according to fish production mode (wild-caught, farm-raised); fish origin (local, imported); food safety information (formalin-free, no safety claim); and the price per kg of whole chilled Rui (BDT 350, BDT 250) (see Table 1). The attributes of fish production mode, food safety information, and fish origin provided information concerning fish traceability. Hence, the information ‘no safety claim’ is the status quo and is the current situation, which is not looking for a change and does not necessarily indicate that the food safety rules were broken for this situation. A focus group was arranged to discuss fish characteristics and decide prices relevant to the local economy.

With the four attributes and two levels, a total of $2^{4}$ (16) hypothetical products can be created. SPSS programming, version 26, provided the minimum number of choice sets from 16 to 4 in the form of a fractional factorial design. Finally, the 404 consumers provided a dataset of *n* = 404 × 3 × 4 = 4848 observations. The choice sets were then randomised and distributed to the participants (see Figure 2).

Before conducting the choice experiments, the respondents read the relevant texts regarding fish attributes (see Appendix A) to reduce the bias resulting from a hypothetical choice experiment [72]. The participants rated all the seven statements on food safety consciousness (see Table 2). The five-point Likert scale items from 1 (no concern) to 5 (very strong concern) showed their perceived value of food safety concerns. A score of two or below represented low safety concerns, three represented average or medium food safety consciousness, and scores of four and above showed high food safety concerns. The participants were asked to define their feelings about food safety concerns by circling one option in each item. Lastly, to gain in-depth insights, consumers’ evaluations of the four fish attributes were determined in a choice ranking survey. Here, agents ranked the fish attributes according to their perceived role of influence in their fish choice from 1 (most influential) to 4 (least influential).

The study employed exploratory factor analysis (EFA) to decide the best number of dimensions and their mutual connotations based on the responses to the particular issues and to build a factor matrix (Table 3). The EFA included all seven statements used to elicit food safety concerns under one dimension, ‘food safety concerns.’ After the completion of the choice experiment, a demographic survey was conducted. In the demographic study, the respondents were asked how much they want to pay for the safe-farmed Rui and safe-wild Rui compared to conventional-farmed Rui and conventional-wild Rui. Finally, consumers’ choice and choice heterogeneity regarding fish attributes and their food safety consciousness were estimated in a CDNL model. Before asking them to participate, the Ethical Review Board, University of Chittagong, Bangladesh, approved the ethical standard of the survey content.

As the study observed the fish choices of those older than 20 and those mainly responsible for family food, the missingness in the sample selection is not random. According to theory, the conditional and multinomial logit model can fit even if the choice is not observed for everyone and if their social status changes frequently. The sample’s unobserved characteristics and non-random sample bias can be adjusted with the Heckman sample-selection model [79]. In this case, ‘unobserved’ means non-measurable factors that may help analyse consumers’ choice regarding the fact that they ‘do most of the fish food shopping for their family’ (see Table 4), specifying the level of responsibility towards the family. The respondents’ choices may differ from those below 20 and who are not responsible for family food. Such differences are also unobserved, and it is uncertain if this unseen sampling may provide biased results. To establish whether there is a shift in the unobserved behaviour regarding households’ fish choice and to control any bias, the study used the Heckman selection model in the STATA program, version 16.

Additionally, socioeconomic variables such as age, gender, number of household members, fish consumption frequency, and percentage of fish bought from supermarkets were measured and detected in the Heckman selection model as independent variables. Furthermore, the two-step procedure of the model was also followed by correcting the non-randomly selected sample bias. The first and second panels in the results show the choice equation and the fish buying (selection) equation. The goodness of fit statistics of the first model report *n* = 4848; Wald χ^2^ (17) = 1546.13; *p*-value (χ^2^) = 0.000; log-likelihood = −2915.862; and non-selected observations = 1236. The results show a positive correlation (ρ) between the residuals of fish choice and fish buying of 0.00045, meaning there is no sample selection bias and indicating that those who are more likely to choose fish are also more likely to buy fish for the family. As the Wald test indicates a non-significant correlation, Heckman’s technique was not used in the main estimated model.

### 3.3. Econometric Model

In economics and marketing, conjoint analysis is widely used to assess consumers’ preferences and demands [80]. In the conjoint valuation model, individuals generally decide to maximise their utility. When they choose an alternative, others are therefore not chosen, indicating that they are mutually exclusive [81]. In the *J* possible alternatives, the utility given by alternative *j* for individual n from the choice set *k* is defined in a linear function as:(1)$$U_{nkj}=x_{kj}^{’}\beta +\varepsilon_{nkj}$$

Hence *β* represents a vector of the significance of the attributes *(x)* for the respondents in evaluating their utility. The error term $\varepsilon_{nkj}\;$ covers the influence of unobserved factors on the utility received by the consumer [81]. The observed part of the utility for an individual is a function of both the product attributes of the possible choices and the characteristics of an individual [82]. Therefore, Equation (2) can be specified as a function of the product attributes and consumer characteristics [83], as follows:(2)$$v_{nj}=\beta_{j}x_{j}+\gamma_{nj}x_{j}z_{n}+u_{nj}$$
where $x_{j}$ is the vector of the fish attributes *j* and $z_{n}\;$ is the vector of characteristics of the individual n. $\beta_{j}\;$ is the utility gained due to the fish attributes *j*, and the model provides for the likelihood of the interactive effects of the fish attributes and consumer characteristics. To analyse consumers’ choice behaviours between several alternatives, it is typical to use the discrete choice model [81]. This is a mathematical function that predicts an individual’s choice based on relative attractiveness or utility [84]. In discrete choice methods, the CNDL model was revealed by McFadden [85] to be consistent with random utility theory. In the choice experiment, respondents had three choices: Option A, Option B, and Option C (buy neither of these). Thus, a conditional logit model is used to estimate the preference [83,86] where the probability of respondent *n* choosing product *j* of choice set *k* can be written as:(3)$$P_{nkj}=\frac{e^{\beta_{j}x_{j}+\gamma_{nj}x_{j}z_{n}}}{\sum_{k}e^{\beta_{j}x_{j}+\gamma_{nj}x_{j}z_{n}}}$$

The algorithm of conditional logistic regression estimates $\hat{\beta }$ for the parameter. Such parameters can be used to analyse the odds of each covariate adjusted for the base cases. The vector is attained by maximising the log-likelihood function, demonstrating that the parameters estimated in the model are applicable for the likelihood of making a choice. Hence, a positive parameter recommends that the independent variable is likely to grow the probability of choosing the particular fish attribute. Alternatively, a negative parameter implies that the predictor value tends to reduce the choice probability. The model outcomes (model 3) provide the estimation results of the model of main effects (product attributes) and main effects with respondent heterogeneity (interactive effects) for the probability of choosing chilled Rui.

Marginal values based on the estimated parameters reflect the WTP for the product attributes. Consumers’ WTP is calculated for by choice modelling (model 3), which is hypothetically assessed. According to Train [81], the estimate can be calculated as the negative ratio of the coefficient of an attribute variable $\left(\beta_{attribute}\right)$ to the price coefficient $\left(\beta_{price}\right)$. The formula is as follows:(4)$$WTP=-\frac{\beta_{attribute}}{\beta_{price}}$$

## 4. Results and Analysis

### 4.1. Descriptive Statistics of Respondents’ Demographics and Socioeconomic Variables

The survey involved a total of 404 households with an average age of 39.65, of which 80.70% were male, 19.10% female, and the remaining 0.20% were reluctant to specify. In Bangladeshi culture, men (almost 80%) are responsible for buying primary food for their family [87]. The average household income of the respondents was BDT 31,7020. (US $1 = BDT 85), and the average monthly household income in Bangladesh is BDT 31,883 [88]. Most of the participants (74.50%) shopped for their family, and 47.80% consumed fish several times per week, indicating that they preferred to buy and consume fish both for themselves and for their families. A total of 84.70% of the participants bought fish from the wet market, whereas the remaining 15.30% bought from supermarkets. This finding is consistent with a recent study by Hoque and Alam [89]. To avoid fish depletion and ensure sustainability, the consumers’ average WTP for safe farmed fish was BDT 299.98/kg, compared to BDT 220/kg for conventionally farmed Rui. In addition, the consumers’ average WTP for safe wild fish was BDT 399.13/kg compared to BDT 350/kg for conventional wild Rui (see Table 4). These two descriptive statistics show that the difference between the mean WTP for traditional and sustainable farmed fish is higher than that between the mean WTP for conventional and sustainable wild fish. The outcomes of a paired samples t-test revealed that the WTP for farmed fish and wild fish were weakly and positively associated (*r* = 0.365, *p* < 0.001) and there was a significant mean difference between the two values of WTP ($t_{4847}$ = −1030.982, *p* < 0.001). The evidence shows that a mean WTP for pale, organically produced salmon is significantly lower than the mean WTP for freedom food salmon [90].

### 4.2. Consumers’ Food Safety Consciousness

The statistics show that the consumers had a high level of consciousness of vitamins, fat and cholesterol, and the use of fish preservatives (e.g., formalin). A total of 39.83% of the respondents had high concern, 31.44% had moderate concern, and 28.73% low concern about the use of preservatives (e.g., formalin) in preserving fish. Bangladeshi consumers are least concerned about additives and artificial colours in fish feed, followed by concerns about genetically modified fish, pesticide residues (toxic chemicals), and microbiological contamination (viruses, fungi); the mean rates of the contents of these are lower than 3, namely 1.89, 2.16, 2.25, and 2.52, respectively (see Table 2, Figure 3). The above results outline that consumers have low consciousness of the hazardous consequences of the use of harmful chemicals in fish. Because of this, traders can easily deceive consumers.

### 4.3. Choice Ranking of Fish Attributes

The importance of fish traceability attributes in fish choice were measured by the contingent valuation method. In doing so, a ranked choice voting system was developed. In this system, consumers ranked four fish traceability attributes by preferences. Based on the outcomes of this ranked choice, we further calculated the relative importance of each attribute, which was measured by the ratio of the range of utility (Rank 1) change of the different attribute levels to the sum of the ranges for all fish product attributes. Figure 4 shows that the ‘production mode’ attribute was the first preference choice for the majority of respondents. As 40% of consumers in Bangladesh distinguish between wild and farmed fish [91], and in general prefer wild fish to farmed fish, this finding can be useful for marketers and policymakers. The claim of safety control (e.g., being formalin-free) was in second place, indicating the second most influential and crucial fish traceability attribute in their choices. Consumers prefer food safety measures, as they are concerned about their health and trust the safety of food produced under government-prescribed standards [92]. Finally, sources of fish origin were considered more important than the price of fresh fish. The relative importance of fish attributes was as follows: production mode (41.09%), safety control information (36.36%), origin of the fish (15.34%), and price (7.20%). 

Finally, as specified in Equation (3), the conditional logit regression was estimated as the impact of the attribute variables on fish choice, with the results reported in Table 5.

### 4.4. Econometric Results

Conditional logit analysis first tests the model fit by examining the Chi-square of the final model. The model’s log-likelihood, the Pseudo R^2^, and the probability of the model likelihood ratio Chi-square are all indicators of a reasonably good fit (see Table 5). Therefore, it can be concluded that the model fits the data. The econometric model provides the fish attributes and interactions between them and the respondents’ socioeconomic variables. Each marginal value in Table 5 represents consumers’ WTP for a particular attribute of a specific fish species, with all else remaining constant.

Typically, consumers who do not have an overall preference for fish as food (opt out) are willing to pay less. They avoid purchasing or consuming either wild or farmed fish as they have little interest in it [93]. In terms of production mode, consumers are less likely to choose farmed fish and are willing to pay less (BDT 119.285/kg) than for wild fish. Consumers’ positive knowledge discrepancy regarding farmed fish is negatively and significantly correlated with their fish choice [89]. Therefore, consumers perceive that farmed fish have lower intrinsic quality in terms of taste and health issues than wild fish [19,20], and consequently are willing to pay less for farmed fish [36].

Consumers have a low preference for imported fish compared to locally produced fish and are willing to pay less for it (BDT 91.428/kg). This finding is consistent with several studies [94,95,96,97]. Consumers prefer formalin-free fish in terms of safety information and are ready to pay a price premium (BDT 124.071/kg) than for fish without such a guarantee (e.g., no safety claim). The finding explains that consumers are concerned about food preservatives, fat and cholesterol, and vitamin content, and are ready to pay more for such labelling which provides safety food information about the product. This finding is also consistent with those of McFadden and Huffman [52] and Suhandoko and colleagues [97], who reported that a meaningful food label and traceability information are crucial for providing information effectively by explaining the relevant attributes associated with the certified production process.

Among the different sociodemographic elements, only consumers’ income has a significant influence on their choice. In general, Rui increases consumers’ utility [69]. Compared to medium-income earners, high-income consumers have less preference for fish consumption and are willing to pay less (BDT 35.571/kg) than those with middle-level income. The existing literature indicates that low-income consumers tend to spend more and rich consumers spend less of their income on fish consumption [98]. When the high-income variable was considered together with the ‘formalin-free’ safety information, the interaction term increased the utility of fish to consumers, meaning that high-income consumers looked for formalin-free and safe fish in the local markets of Bangladesh. They were willing to pay a price premium of BDT 71.785/kg for formalin-free fish. Moreover, highly educated consumers were also more likely to prefer fish labelled with safety information such as being formalin-free. They are willing to pay a price premium for such fish. The findings suggest that consumers with a high level of income and education prefer food quality to food quantity [99].

Individually, consumers with low food safety concerns had a positive marginal WTP. However, when such low concerns are considered ‘no safety claim’, in a status-quo state, the results show that they decrease the utility of fish for consumers, meaning that they are substitutes. Because of this substitute effect, consumers are willing to pay less for unchecked fish, meaning that even low food safety-conscious consumers prefer safe fish, or fish with safety information because serious diseases such as cancer result from the long-term formalin consumption from adulterated fish [100]. On the other hand, high-safety-concern consumers’ marginal WTP is negative. However, the interaction term between high safety concern and ‘formalin-free’ is positive and significant, meaning that they are complementary, and because of this positive, complementary effect, high-safety-conscious consumers are more likely to prefer formalin-free fish. They are also willing to pay a price premium of BDT 34.43 for such fish. The findings imply that labelling which includes information about formalin-free fish encourages consumers to buy fish at higher prices. The findings are also supported by McFadden and Huffman [52], who explain that information about the certified cultivation of food motivates consumers to pay a price premium.

## 5. Discussion

Consumers in Bangladesh are concerned about vitamins, fat/cholesterol, and the preservatives (e.g., formalin) used to preserve chilled fish. Many food scandals and an integrated approach to food safety control taken by non-governmental organisations (NGOs) and government authorities have made consumers become safety conscious. In addition, consumers are less concerned about additives and artificial colours in fish feed, genetically modified fish, pesticide residues (toxic chemicals), and microbiological contamination (virus, fungi) in chilled fish. They have little concern about these aspects because of their limited knowledge of critical safety violations concerning food handling. They also underestimate foodborne diseases related to microbial contamination and have low personal hygiene. Consumers’ lack of knowledge of food adulteration and their low concerns of the risky consequences of using harmful chemicals in fish production means traders can deceive consumers easily. Therefore, consumer education can help to reduce foodborne diseases, and they need to make informed safety-conscious seafood purchasing decisions.

When choosing fish for consumption, consumers mainly rely on their production mode (wild or farmed fish), followed by their food safety concerns. In addition, some consumers do not have a preference for fish (i.e., farmed or wild) and are willing to pay less for fish as they have limited consciousness of the selling of fish and its cultivation process [101]. However, consumers who prefer fish prefer local sources of fish origin and are willing to pay more for it than for imported fish. First, consumers can easily find out about the local fish cultivation process compared to that of imported fish. Second, they are willing to pay less for imported fish as they are less likely to prefer it due the transportation distances involved [102] and their trust in domestic food standards [103,104]. Third, they perceive local fish to be fresher than imported fish, as the local fish supply chain from producer to consumer involves less time than imported fish. Such considerations motivate consumers to buy local fish rather than imported ones.

Consumers are less likely to prefer farmed fish over wild fish. The findings indicate that consumers are currently more concerned about different food sensory attributes such as nutritional value, taste, and health issues [105], and consequently perceive farmed fish to be inferior to wild fish [17,38]. Another reason for not consuming, or consuming less, farmed fish is the high frequency of formalin usage that causes health hazards [29]. Such a situation indicates the necessity of food traceability, which can create trust among consumers about fish consumption and ensure fish procurement. Moreover, consumers with an awareness of safety food information want to consume formalin-free fish compared to fish with no safety guarantees, which indicates that they have great interest in safety food information. In this case, labelling food can play a crucial role in providing information and building trust among consumers regarding food safety and food quality. Such labelling can provide food traceability for food safety in terms of tracking information related to the food supply chain from farms to consumers.

In Bangladesh, even consumers with low food safety concerns are less likely to prefer fish without a guarantee that they are free from excessive food additives such as formalin. Such findings indicate that even consumers who are less concerned about food safety are looking for food safety information and safe fish compared to unguaranteed fish with no safety claims. High-income consumers in Bangladesh are less likely to prefer fish and are willing to pay less for it. When the formalin-free information was provided to high-income people, they were most likely to choose the whole fish. Furthermore, although not statistically significant, highly educated and high-food-safety-conscious consumers are also less likely to prefer fish. Such findings show that highly knowledgeable consumers with high incomes are more aware of food contamination in Bangladesh [106,107], whereas consumers with low concerns and low incomes have limited knowledge of the food production process; however, they are willing to pay less for fish without safety checks or safety guarantees. Besides, highly educated, high-income, and very food-safety-conscious consumers are more likely to prefer additive-free or safe fish and are willing to pay a price premium for such products.

The study has researched consumers’ food safety consciousness and their preferences and readiness to pay a premium for fish traceability attributes, especially regarding chilled Rui in Bangladesh. For this purpose, data were collected from urban households of Bangladesh in which 47.80% of respondents ate fish several times a week, indicating that people in Bangladesh prefer to purchase and consume fish for themselves and their families, as 74.50% of the respondents went shopping for their family. To improve fish procurement and avoid fish depletion, consumers are willing to pay higher prices for safe farmed fish than for safe wild fish. However, on average, consumers have a lack of concern about checking for food adulteration, nor are they very conscious of the dangerous situation emerging from the use of harmful chemicals in fish and additives and artificial colours in fish feed, genetically modified fish, pesticide residues (toxic chemicals), or the microbiological contamination (virus, fungi) of fish. Therefore, a more engaging means of communication and an education enhancement programme will help to make consumers informed and safety conscious.

## 6. Conclusions

Government agencies and NGOs should focus on labelling, including detailed information from farm to fork. Such labelling information could play a crucial role in a food traceability program to build trust among consumers about food safety and quality. Furthermore, the HACCP and Codex standards should be incorporated into the existing diverse regulations to prevent the adulteration of fish and fish products. Then, the regular surveillance, inspection, and random sampling of fish by concerned food safety officers of the state could contribute to the institutionalisation and good governance of fish and fish products control systems in value chains. Increasing the safety standards and levels of product knowledge will improve consumer protection from deceptive trading and change consumers’ attitudes, resulting in fish buying decisions.

Consumers who do prefer fish are interested in local fish and are ready to pay more for them than for imported fish, as they appreciate their freshness compared to imported fish. At present, consumers are less likely to prefer farmed fish and are prepared to pay less for them than for wild fish. They are more sensitive to health issues resulting from consuming fish containing formalin and other additive and to different food sensory attributes such as nutrition value and taste, and thus they consider farmed fish to be inferior to wild fish. Furthermore, consumers who possess food safety information want to consume formalin-free fish instead of fish with no guarantees. Amongst low-education, low-income consumers who have little concern for food safety, their preferences towards fish are positive, and they are prepared to pay a price premium. However, they are willing to pay less for ‘no safety claimed’ fish. On the other hand, highly educated, high-income consumers with high food-safety-concerns have less preference for fish consumption in general and are willing to pay less for them, because that they are more concerned about food contamination in Bangladesh. They are thus more likely to eat formalin-free fish and are ready to pay a price premium. Overall, regardless of low or high income, education, or safety awareness, everybody is looking for safe fish with adequate and explicit safety and traceability information. Therefore, locally produced fish can be marketed as having been supplied in a way that provides a formalin-free label, and then producers and marketers can impose a price premium for their fish products.

The study’s main contribution concerns consumers’ consciousness of food traceability information in the significant seafood market of an emerging economy. Considering such an emerging market (e.g., Bangladesh) in a traditional value chain, exploring consumers’ perceptions of fish traceability attributes is vital to estimating the demand for safe fish and fish products. Bangladeshi consumers prefer to eat safe fish, with traceability information helping to indicate the safety of fish. Therefore, to meet the increased fish consumption demand with food safety controls, policymakers, marketers, and producers should focus on food traceability programs, as consumers prefer formalin-free fish. These initiatives will provide information to consumers about the quality and health benefits of fish (i.e., better safety and fewer chemical residues) and significantly improve the WTP for such products. Consequently, the traceability information could optimise the production process and supply chain and help make seafood recall management more effective.

The study may have attention bias as it follows a choice experiment approach in which participants were requested to select from a given number of fish attributes and different levels of them. Therefore, in future research, other valuation approaches such as auctions or real choice experiments could be used to measure choices and WTP. Another limitation of the study is its data collection, which only focused on Dhaka and Chittagong. Therefore, it is not easy to ensure that the study sample represents all Bangladeshi consumers’ concerns because of variances in economic development, education levels, and food consumption habits across the country. More classification of fish and fish products (coastal and marine version, whole and fillet) and Bangladesh’s all divisions could be considered in future research, representing potential fish markets with food traceability systems to elicit better results. Despite the limitations, this study could be the base for conceptualising other studies that aim to be related to other food traceability attributes and food safety programs.

## Figures and Tables

**Figure 1 foods-11-01675-f001:**
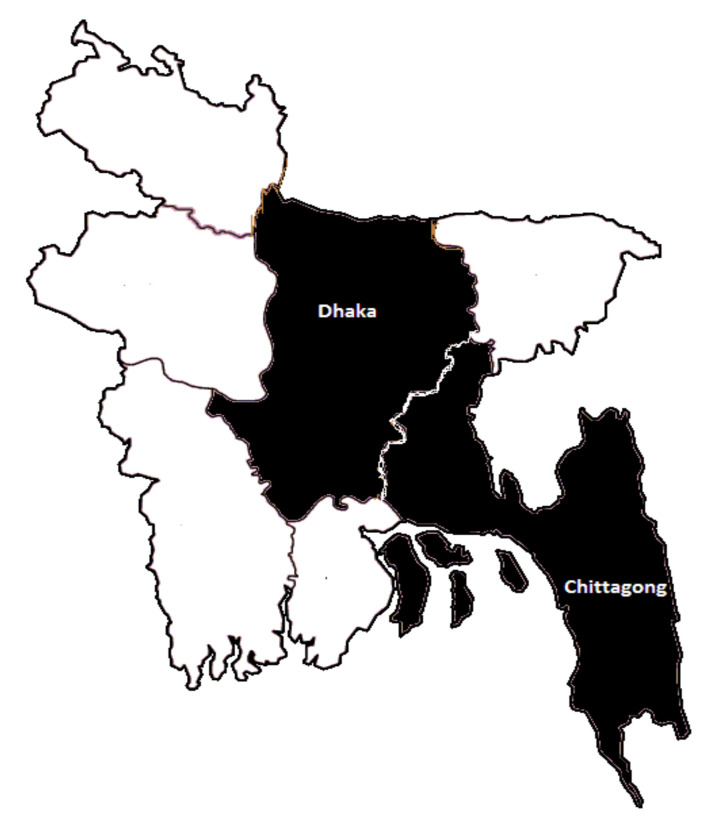
The black shading indicates the study area.

**Figure 2 foods-11-01675-f002:**
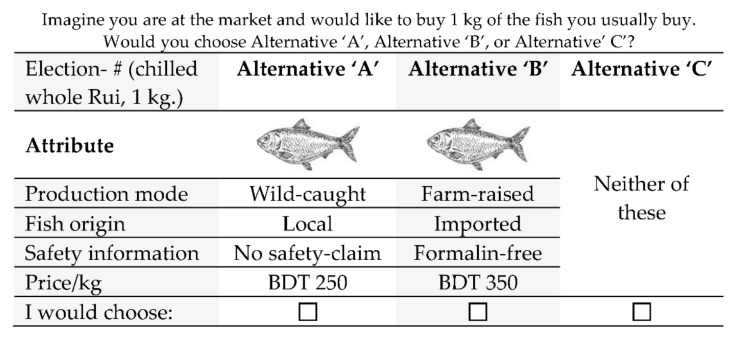
Example of a choice set. # = 1, 2, 3, 4.

**Figure 3 foods-11-01675-f003:**
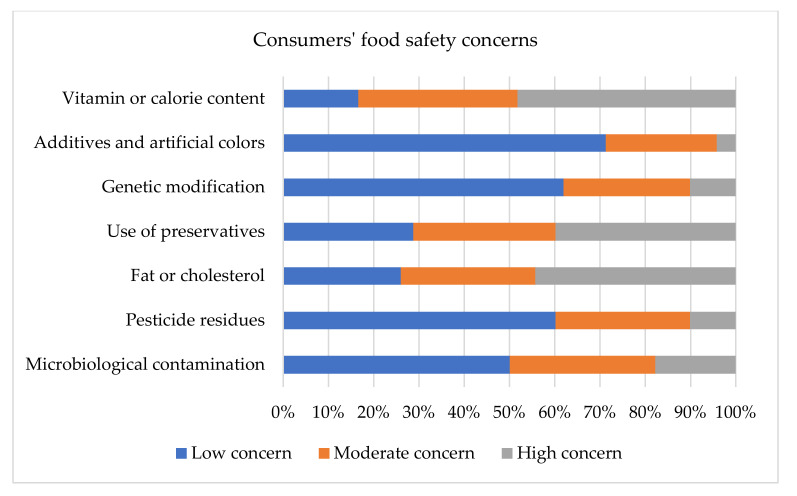
Consumers’ safety concerns toward fish were identified based on five-point Likert scaling where the horizontal axes indicates the percentage of respondents. Respondents’ scores of 2 or below were regarded as a low concern. Those who gave scores 3 were deemed to be a moderate concern. Lastly, scores of 4 and above indicate their high concern. The vertical axes measures concern type.

**Figure 4 foods-11-01675-f004:**
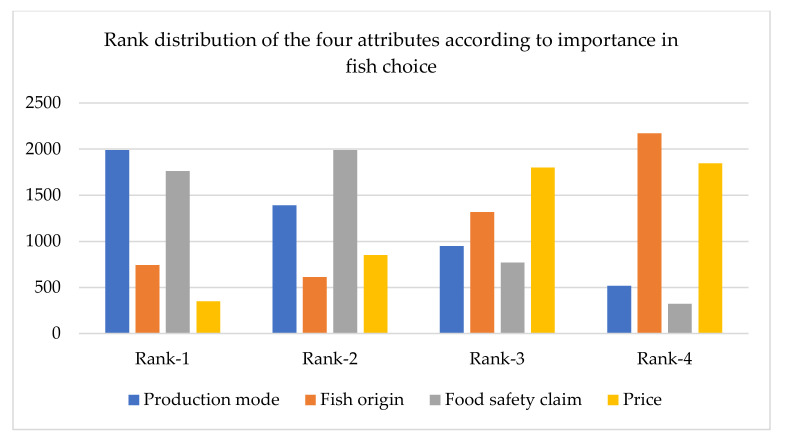
Consumers fish attribute choice. The vertical axes indicates the number of observations.

**Table 1 foods-11-01675-t001:** Fish attributes and levels for the choice experiments.

Fish Attribute	Description	Levels
Production mode	Wild fish are caught at sea or in rivers, lakes, and other natural water bodies, while farmed ones are raised in fresh inland water or coastal areas in brackish or marine saline water.	-Wild-caught -Farm-raised
Fish origin	We can cultivate and explore fish domestically or import fish cultivated in foreign countries (e.g., Burmese Rui/Indian Rui).	-Local -Imported
Safety (control) information	Local government regulatory activity provides consumer protection and ensures that fish are free from formalin and safe for consumption. However, no authorised body guarantees that fish during storage, processing, and distribution are free from formalin and safe for consumption.	-Formalin-free -No safety claim (status quo)
Price	This is the cost of purchase—what consumers would pay for one kg of Rui fish. Here, it is denoted in the Bangladeshi currency, taka, globally coded as BDT.	-BDT 250/kg -BDT 350/kg

**Table 2 foods-11-01675-t002:** Descriptive statistics of consumers’ food safety concerns derived from the elicitation study.

Sl	Food Safety Observations and References	Mean and S.D. of Scores	Rank
a.	I am concerned about the microbiological contamination of chilled fish.	2.52 ± 0.986	4
b.	I am concerned about pesticide residues (toxic chemicals) in chilled fish.	2.25 ± 0.992	5
c.	I am concerned about the fat or cholesterol content of chilled fish.	3.30 ± 1.058	2
d.	I am concerned about the use of preservatives to preserve chilled fish.	3.19 ± 1.035	3
e.	I am concerned about genetically modified fish.	2.16 ± 1.031	6
f.	I am concerned about additives and artificial colours in fish feed.	1.89 ± 0.927	7
g.	I am concerned about the vitamin content of chilled fish.	3.46 ± 0.973	1

N = 404; S.D. = Standard deviation; 1 = “not concerned” to 5 = “very strongly concerned”. a. = [73]; b. = [74]; c. = [75]; d. = [29]; e. = [76]; f. = [77]; and g. = [78].

**Table 3 foods-11-01675-t003:** Exploratory factor analysis outcome.

Sl.	Observed Variables	Food Safety Concerns
		Factor Loadings
1.	I am concerned about the microbiological contamination (virus, fungi) of chilled fish.	0.838
2.	I am concerned about pesticide residues (toxic chemicals) in chilled fish.	0.841
3.	I am concerned about the fat or cholesterol content of chilled fish.	0.887
4.	I am concerned about the use of preservatives (e.g., formalin) to preserve chilled fish.	0.869
5.	I am concerned about genetically modified fish (genetically altered using genetic engineering).	0.827
6.	I am concerned about additives and artificial colours in fish feed.	0.809
7.	I am concerned about the vitamin or calorie content of chilled fish.	0.882
	Eigenvalue	5.069
	KMO score	0.910
	Bartlett’s test of sphericity: approximate Chi-square (*χ^2^*)	27,279.880
	Degrees of freedom (*d.f.)*	21.000
	Total variance explained (%)	72.412
	Determinant of the correlation matrix	0.004
	Cronbach’s Alpha (α) (*n* = 7)	0.936

**Table 4 foods-11-01675-t004:** Descriptive statistics of the demographic, and psychographic variables and the preference patterns for chilled Rui.

Sample Size (Households)	404
Age (mean ± S.D.)	39.65 ± 9.91
Gender (%)	
Male	80.70
Female	19.10
Do not want to specify	0.20
Years of education (mean ± S.D.)	15.16 ± 2.79
Household monthly income (’000) (mean ± S.D.)	317.02 ± 16.72
Profession (%)	
Employed	72.80
Self-employed	18.80
Housemaker	7.70
Pensioner	0.70
Do you do most of the fish food shopping for your family? (%)	
Yes	74.50
No	25.50
Overall fish consumption (%)	
Less than once per month	9.90
Once per month	3.20
Once per week	35.70
Several times per week	47.80
Daily	3.50
Where fish bought from? (%)	
Wet market	84.70
Supermarket	15.30
Percentage of fish that consumers buy from supermarkets (mean ± S.D.)	18.77 ± 27.04
WTP for safe-farmed Rui (conventional-farmed Rui is BDT 220/kg)	299.98 ± 5.73
WTP for a safe-wild Rui (conventional-wild Rui is BDT 350/kg)	399.13 ± 6.13
*n* = 4848	

The monthly income is calculated in the Bangladesh local currency, the Taka, which is globally coded as BDT; USD 1 = BDT 85.

**Table 5 foods-11-01675-t005:** Conditional choice model estimate with fish attributes.

Variables	Choice of Chilled Rui in the Conditional Logit (CNDL) Model				
	Model (1) with Fish Attributes and Socioeconomics Variables	Model (2) with Fish Attributes, Socioeconomics Variables, and Their Interactions	Consumers’ WTP Based on the Model (2) for Fish Attributes, Socioeconomic Variables, and Their Interactions		
	Coefficient	Coefficient	WTP	S.E.	C.I.
Farmed fish	−1.659 *** (0.090)	−1.670 *** (0.090)	−119.285	11.623	[−143.809, −94.762]
Imported fish	−1.270 *** (0.125)	−1.280 *** (0.125)	−91.428	14.029	[−121.028, −61.828]
Formalin-free	2.472 *** (0.132)	1.737 *** (0.257)	124.071	19.154	[83.659, 164.483]
Price	−0.014 *** (0.001)	−0.014 *** (0.001)	--	--	--
Opt out	−6.754 *** (0.356)	−7.036 *** (0.389)	−502.571	22.987	[−551.071, −454.071]
Consumers’ low FSC	0.003 (0.092)	0.124 (0.113)	8.857	8.063	[−8.154, 25.869]
Consumers’ high FSC	0.003 (0.120)	−0.231 (0.166)	−16.500	11.952	[−41.716, 8.716]
Low income	0.001 (0.092)	0.017 (0.113)	1.214	8.091	[−15.857, 18.286]
High income	−0.005 (0.134)	−0.498 *** (0.192)	−35.571	14.061	[−65.238, −5.904]
Low education	−0.001 (0.208)	0.015 (0.251)	1.071	17.952	[−36.805, 38.948]
High education	0.001 (0.098)	−0.201 (0.130)	−14.357	9.377	[−34.141, 5.426]
Low FSC * No safety claim		−0.336 * (0.178)	−24.000	12.865	[−51.143, 3.143]
Low education * No safety claim		−0.048 (0.414)	−3.428	29.574	[−65.824, 58.967]
Low income * No safety claim		−0.045 (0.182)	−3.214	13.031	[−30.707, 24.279]
High education * Formalin-free		0.428 ** (0.185)	30.571	13.446	[2.202, 58.940]
High income * Formalin-free		1.005 *** (0.264)	71.785	19.721	[30.176, 113.394]
High FSC * Formalin-free		0.482 ** (0.228)	34.428	16.521	[−0.428, 69.285]
N = 4848; Group = 404	Pseudo-R2 = 0.2968, LR Chi^2^ (11) = 1822.71, probability (Chi^2^) = 0.000	Pseudo-R^2^ = 0.3016, LR Chi^2^ (17) = 1852.55, probability (Chi^2^) = 0.000			

Standard errors in parentheses; *** *p* < 0.001, ** *p* < 0.05, ND * *p* < 0.1. Parameter estimates from the CNDL model; FSC = Food Safety Concern. The WTP, standard errors (S.E.), and confidence intervals (C.I.) were estimated with the Delta method.

## Data Availability

The data are available on request from the corresponding author.

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
