# Peer review of "Consumers’ Preferences for the Traceability Information of Seafood Safety"

_foods, 2022, doi:10.3390/foods11121675_

Round 1
Reviewer 1 Report
Thank you very much for giving me the opportunity to read the manuscript Consumers' preferences for the traceability information of seafood safety. The topic is interesting and relevant for special issue “Recent Advances in Consumers’ Preferences and Behavior toward Healthy and Functional Foods.
The research is very interesting to read, however some critical issues should be solved.
Abstract
Line 27 I suggest to use third person instead of first person
Literature
I think that literature should be partially enriched, since there are many studies reporting similar methodological approaches:
Olesen, I.; Alfnes, F.; Røra, M.B.; Kolstad, K. Eliciting consumers’ willingness to pay for organic and welfare-labelled salmon in a non-hypothetical choice experiment. Livest. Sci. 2010, 127, 218–226;
Davidson, K.; Pan, M.; Hu, W.; Poerwanto, D. Consumers’ willingness to pay for aquaculture fish products vs. wild-caught seafood—A case study in Hawaii. Aquac. Econ. Manag. 2012, 16, 136–154
I suggest also to highlight the importance of sustainability in the fish consumer behaviuor by citing the following papers:
Zhou, G.; Hu, W.; Huang, W. Are consumers willing to pay more for sustainable products? A Study of eco-labeled tuna steak. Sustainability 2016, 8, 494
Maesano, G., Di Vita, G., Chinnici, G., Pappalardo, G., & D'Amico, M. (2020). The role of credence attributes in consumer choices of sustainable fish products: A review. Sustainability, 12(23), 10008.
Finally, I recommend to cite the following studies on perceived quality and the price-quality relationship in important coastal areas
Olsen, S. O., Tuu, H. H., & Grunert, K. G. (2017). Attribute importance segmentation of Norwegian seafood consumers: The inclusion of salient packaging attributes. Appetite, 117, 214-223.
Wang, O., & Somogyi, S. (2018). Chinese consumers and shellfish: Associations between perception, quality, attitude and consumption. Food quality and preference, 66, 52-63.
Conclusion
I think that authors should put more efforts into providing further insights for marketers and producers implications
Author Response
June 01, 2022
Reference: "Review: " Consumers' preferences for the traceability information of sea-food safety" at foods."
Dear Reviewer,
As the corresponding author, I am sending the rebuttal letter explaining the changes made to the manuscript. The changes have been made on your comments, where we found the comments are helpful and constructive to fulfil the journal's criterion and give the scientific discourse. The revision starting with key and relevant points are addressed below.
In the Abstract, Line 27, I suggest to use third person instead of first person.
Yes, according to the reviewer's suggestion, the first person has been replaced by the third person. Please see line 27 in the abstract of the paper.
I recommend to cite the relevant studies on perceived quality and the price-quality relationship in important coastal areas.
Many thanks to the reviewers for this good observation. As per reviewer recommendations, the most relevant study by Olesen and colleagues (2010) [Olesen, I.; Alfnes, F.; Røra, M.B.; Kolstad, K. Eliciting consumers' willingness to pay for organic and welfare-labelled salmon in a non-hypothetical choice experiment. Livest. Sci. 2010, 127, 218–226) is cited now. Please see line 401.
In Conclusion, I think that authors should put more efforts into providing further insights for marketers and producers' implications.
We agree with the reviewer's comment; based on the findings, additional insights are provided for producers' and marketers' policy implications. Please see the line 589-591.
Reviewer 2 Report
"This paper introduces a new food traceability attribute, the safety claim of being ‘formalin free,' and status quo 'no safety claim' to obtain consumers real insights into the traceability information of food safety in fish consumption. The authors explore seafood markets by assessing the demand for traceability information attributes, utilizing data from an experimental survey in Bangladesh. The authors gave an effort presenting literature review, data collecting and the analyses performed. The results are presented in satisfied form, and are followed with tables and figures that gives the readers feasibility to follow the paper. The study is relevant and interesting in summarizing the information regarding fish consumption and consumer habits, and food traceability systems, and attributes. The English quality is satisfied, and some fine spelling/grammar checks should be performed. The authors answered to all aims of the study and indicated some disadvantages of the study, e.g. data collection based on Dhaka and Chitagong. On the base of this study government agencies and NGOs should focus on labelling, including detailed information from the fish cultivation process to consumption. This information can play a crucial role as a food traceability program to build trust among consumers about food safety and quality. This study could be the base for the conceptualization of the other studies that aim is related to the other food traceability attributes and food safety programs. Conclusion – it must be more clear, shorter, and not summarizing whole study in short. Please conclude directly answering to your aims."Author Response
June 01, 2022
Reference: "Review: " Consumers' preferences for the traceability information of sea-food safety" at foods."
Dear Reviewer,
As the corresponding author, I am sending the rebuttal letter explaining the changes made to the manuscript. The changes have been made on your comments, where we found the comments are helpful and constructive to fulfil the journal's criterion and give the scientific discourse. The revision starting with key and relevant points are addressed below.
On the base of this study, government agencies and NGOs should focus on labelling, including detailed information from the fish cultivation process to consumption. This information can play a crucial role as a food traceability program to build trust among consumers about food safety and quality. This study could be the base for the conceptualization of the other studies that aim is related to the other food traceability attributes and food safety programs. Conclusion – it must be more clear, shorter, and not summarizing whole study in short. Please conclude directly answering to your aims."
We do agree with the reviewer's comment, and as per the recommendation, now the section Conclusion is made shorter, deleting the part 'summarising the whole study in short'. Now this section received focuses on answering the aims of to study. Please see lines 566-570.
Reviewer 3 Report
This study aims to investigate seafood markets by assessing the demand for traceability information attributes, using survey data among Bangladeshi citizens. The results show that citizens pay attention to vitamins, cholesterol and preservatives, while they are little concerned about microbiological contamination, pesticide residues, genetic modifications and additives. The introduction is satisfying and explanatory, although especially in the initial sentences the English and the form are very elementary: please improve these part.
The experimental design and the results are well described and discussed.
551-561 these considerations should be included in the discussion of the results and not in the conclusions.
I asked to move a part of the conclusion (Lines 551-561) in the discussion, and change with less elementary (for a scientific text) english of the sentences in lines 33-36.
Author Response
June 01, 2022
Reference: "Review: " Consumers' preferences for the traceability information of sea-food safety" at foods."
Dear Reviewer,
As the corresponding author, I am sending the rebuttal letter explaining the changes made to the manuscript. The changes have been made on your comments, where we found the comments are helpful and constructive to fulfil the journal's criterion and give the scientific discourse. The revision starting with key and relevant points are addressed below.
I asked to move a part of the Conclusion (Lines 551-561) in the Discussion, and change with less elementary (for a scientific text) English of the sentences in lines 33-36.
Many thanks to the reviewer for this constructive suggestion. As per the comment, line 551-562 has been transferred from section Conclusion to the Discussion. Lastly, in lines 33-36, English of the sentence has been changed based on scientific writing.